# The Effect of Pre-Stretched Substrate on the Electrical Resistance of Printed Ag Nanowires

**DOI:** 10.3390/nano13040719

**Published:** 2023-02-13

**Authors:** Yoon Jae Moon, Chaewon Kim, Eunsik Choi, Dong Yeol Shin, Kyung-Tae Kang

**Affiliations:** Digital Transformation R&D Department, Korea Institute of Industrial Technology, Ansan 15588, Republic of Korea

**Keywords:** silver nanowires, pre-stretching, printing, stretchable electrode, sliding, resistance

## Abstract

One-dimensional nanomaterials have drawn attention as an alternative electrode material for stretchable electronics. In particular, silver nanowires (Ag NWs) have been studied as stretchable electrodes for strain sensors, 3D electronics, and freeform-shaped electronic circuits. In this study, Ag NWs ink was printed on the pre-stretched silicone rubber film up to 40% in length using a drop-on-demand dispenser. After printing, silicone rubber film was released and stretched up to 20% as a cyclic test with 10-time repetition, and the ratios of the resistance of the stretched state to that of the released state (R_stretched_/R_released_) were measured at each cycle. For Ag NWs electrode printed on the pre-stretched silicone rubber at 30%, R_stretched_/R_released_ at 10% and 20% strain was 1.05, and 1.57, respectively, which is significantly less than about 7 for Ag NWs at the 10% strain without pre-stretched substrate. In the case of 10% strain on the 30% pre-stretched substrate, the substrate is stretched and the contact points with Ag NWs were not changed much as the silicone rubber film stretched, which meant that Ag NWs may slide between other Ag NWs. Ag NWs electrode on the 40% pre-stretched substrate was stretched, strain was concentrated on the Ag NWs electrode and failure of electrode occurred, because cracks occurred at the surface of silicone rubber film when it was pre-stretched to 40%. We confirmed that printed Ag NWs on the pre-stretched film showed more contact points and less electric resistance compared to printed Ag NWs on the film without pre-stretching.

## 1. Introduction

Typical applications of flexible and stretchable electronic devices include stretchable displays [1,2,3,4], skin sensors of pressure [5,6], electronic textile [7], and muscle-like soft actuators [8]. Electrode of the stretchable and flexible electronic device should maintain the initial electrical conductivity in despite of various deformations. One-dimensional nanomaterials have drawn attention as an alternative electrode material for stretchable electronics [9,10,11,12]. Especially, silver nanowires (Ag NWs) have been studied as stretchable electrodes materials for strain sensors [13], 3D electronics [14] and freeform-shaped electronics [15], and printed circuit boards [16,17].

Ag NWs electrode for stretchable electronics have been made using diverse coating [18,19] and printing methods [20,21,22]. The drop-on-demand (DOD) printing method using a dispenser has several advantages, such as preventing nozzle clogging due to the high aspect ratio of Ag NWs and wider ink viscosity range, compared with the inkjet printing method. Additionally, data for floor plans saved in computers is directly transferred to a dispenser printer as fully digitalized manufacturing process, and the pattern is embodied.

In order to avoid failure of electrodes upon stretching, the dominant approach is making electrodes wavy or buckled shape [23,24]. The alternative approach is using the networks of one-dimensional nanomaterials, which are highly conductive and stretchable conductive materials. In order to increase the contact points with nanomaterials, stretchable elastomers are pre-stretched before formation of conductive components [25,26].

Currently, a method of printing or coating in a pre-stretched state is widely used when forming a silver nanowire electrode on a stretchable substrate. However, a few extensive research projects on physical behavior of Ag NWs on the elastic substrate with the continuous electrical resistance measurement have been performed as Ag NWs printed on pre-stretched substrate was stretched and released [27,28]. In this study, Ag NWs line was printed on the pre-stretched silicone rubber film using DOD dispenser with various pre-stretching percent of silicone rubber film. After sintering, silicone rubber film was released and stretched up to 20% as a cyclic test with 10-time repetition. While applying tensile strain, electrical resistance was measured in situ. The effect of the pre-stretching substrate on the electrical resistance of printed Ag NWs was further investigated in detail.

## 2. Experiment

The composition of Ag NWs ink (Flexioink, SG Flexio, Daejeon, Republic of Korea) consisted of 0.14 wt% Ag NWs (20 nm diameter, 20 µm length), dispersed in deionized water. Recommended sintering temperature is 120 °C for 30 min. Ag NWs ink was printed on the pre-stretched silicone rubber film (thickness 300 μm) using the DOD dispenser (MDV3200A, Vermes, Plain city, OH, USA), based on piezo technology. The nozzle diameter of dispenser was 150 µm. The pre-stretching percent was varied from 10% to 40%. The experimental procedures is as follows. As shown in Figure 1a, silicone rubber film was stretched at first using automatic stretching equipment (Sciencetown, ST1, Incheon, Republic of Korea). Before pre-stretching, UV/O3 treatment was performed to change the hydrophobic surface to hydrophilic surface for 25 min. After stretching, stretched silicone rubber film was fixed using the fixing jig, which was custom made. Ag NWs line (20 mm in length, 1.1 mm in width) was printed on silicone rubber film on the fixing jig. Additionally, printed Ag NWs line was dried and sintered on the fixing jig at 120 °C for 30 min (Figure 1b). Liquid metal (EGaIn, Sigma-Aldrich, St. Louis, MO, USA) and copper wire were used as electrical pad to measure the electrical resistance in situ. In order to fix the liquid metal and copper wire during stretching test, liquid metal and copper wire were covered with Polydimethylsiloxane (PDMS), and then curing PDMS was performed at 70 °C for 20 min (Figure 1c). After curing, pre-stretched silicone rubber film was released. Released silicone rubber film was stretched to 10% and 20% as a cyclic test with 10 time repetition (Figure 1d). During a cyclic test with 10-time repetition, the electrical resistance was measured in situ using multi-meter (8540A Fluke, Everett, WA, USA) at intervals of one second. The electrical resistance of Ag NWs was related to the change of morphology. In order to observe morphology, change of Ag NWs, FESEM (Hitachi, SU8010, Tokyo, Japan) measurement was carried out.

## 3. Results

Figure 2 shows the in situ resistance change of Ag NWs electrode on variously pre-stretched silicone rubber film during a cyclic test with 10 times repetition applying 10% strain. As pre-stretching of silicone rubber film percent increases, the ratios of the resistance of the stretched state to that of the released state (R_stretched_/R_released_) at first cycle decrease up to pre-stretching 30%. Figure 3 shows the crack formation at the surface of silicone rubber film at 0% strain and cutting of Ag NWs at 10% strain in the case of 40% pre-stretching. It is thought that at pre-stretching 40%, this resulted in the increase of the ratio of R_stretched_/R_released_. The ratios of the resistance of R_stretched_/R_released_ at pre-stretching 10, 20, 30, and 40% were 2.02, 1.22, 1.05, and 1.1, respectively. At pre-stretching 30%, the ratio of R_stretched_/R_released_ was minimum.

As applying 20% strain, the resistance change of Ag NWs electrodes on pre-stretched silicone rubber film during a cyclic test with 10 times repetition is shown in Figure 4. As shown in Figure 2, the ratios of the resistance of the stretched state to that of the released state (R_stretched_/R_released_) at first cycle decrease up to pre-stretching 30%. At pre-stretching 40%, the ratio of R_stretched_/R_released_ increases. However, at pre-stretching 30 and 40%, failure of Ag NWs electrode occurred at 9th cycle and 4th cycle, respectively. The strain was concentrated on Ag NWs electrodes due to cracks in silicone rubber film formed during pre-stretching the cracks of silicone rubber film was formed [28]. The ratios of the resistance of R_stretched_/R_released_ at pre-stretching 10, 20, 30, and 40% was 5.88, 3.35, 1.57, and 2.04, respectively. Likewise, 10% strain, the ratios of the resistance of R_stretched_/R_released_ at pre-stretching 30% was the minimum.

Figure 5 represents the ratio change of R_stretched_/R_released_ of Ag NWs electrode with pre-stretching percent of silicone rubber film at first cycle. At pre-stretching 10%, the value of R_stretched_/R_released_ applying 10% strain was approximately 3 times higher than that of R_stretched_/R_released_ applying 20% strain. As increasing pre-stretching strains, the difference of R_stretched_/R_released_ applying 10 and 20% strain decreased. At pre-stretching 30%, the value of R_stretched_/R_released_ was minimum. At pre-stretching 40%, R_stretched_/R_released_ began to increase. In the case of R_stretched_/R_released_ of Ag NWs without pre-stretching was 7 at 10% strain. The resistance of Ag NWs was not changed much in the stretched state.

Figure 6 shows the cross sectional FESEM images of Ag NWs electrodes. In Figure 6a, the wavy structure of Ag NWs electrode was not shown at 10% pre-stretching. At 0% strain, the wavy structure began to be shown from 20% pre-stretching. From 30% pre-stretching, Ag NWs electrode had the clear wavy structure. The number of Ag NWs with pre-stretching increased. The contact point with Ag NWs increased at pre-stretched silicone rubber film. As shown in Figure 6b, at 10% pre-stretching, Ag NWs electrode are not changed much after 10 and 20% cyclic stretching test. However, Ag NWs electrode printed on the 30 and 40% pre-stretched silicone rubber film was separated from the silicone rubber film. As shown in Figure 3, in the case of pre-stretching 40%, the silicone rubber film had the crack at 0% strain. The crack resulted from the formation of silicon oxide by UV/O_3_ treatment to change the hydrophobic surface of silicone rubber film to the hydrophilic surface [29]. At 40% pre-stretching, a wavy pattern was observed at the surface of silicone rubber film. These wavy patterns are thought to have been caused by the surface of silicone rubber film that was partially damaged when pre-stretched. Therefore, when tensile strain is repeatedly applied, cracks are formed in this part, and it is thought that the Ag NWs electrode is broken in the tensile strain of more than 20%. In Figure 7, Ag NWs was bent. At pre-stretching 40%, Ag NWs are cut off after 10% cyclic stretching test, which red rectangle part is enlarged. At 20% strain, the contact point between printed Ag NWs on 40% pre-stretched silicone rubber film was reduced as shown in Figure 7.

It was reported that the reason, why the resistance of the nanowire electrode formed on the pre-stretched substrate did not differ significantly during tensile strain being applied and removed, resulted from the reversible sliding of nanowire [27]. Figure 8 showed the schematics that Ag NWs movement when that Ag NWs ink was printed, dried, and sintered on pre-stretched silicone rubber film. When printed Ag NWs electrode was released, the contact point with Ag NWs increased. As printed Ag NWs electrode was stretched at low strain, the number of contact point with Ag NWs at stretched and released state is maintained by reversible sliding of Ag NWs. However, as printed Ag NWs electrode was stretched at high strain, the number of contact point with Ag NWs at stretched state decreased compared to the number of contact point with Ag NWs before the stretching test by sliding and rotation of Ag NWs, which was due to the poor adhesion between Ag NWs and silicone rubber film [30]. As strain increased, the resistance of Ag NWs electrode at stretched state increased. As the number of cyclic increased, the number of contact point between Ag NWs is also gradually reduced due to the rotation and sliding of Ag NWs, as shown in Figure 7. It resulted in gradual increase of resistance of printed Ag NWs electrode at released and stretched state.

## 4. Summary

Ag NWs electrode was printed on the various pre-stretched silicone rubber film using DOD dispenser. The difference of R_stretched_ and R_released_ of Ag NWs printed on the pre-stretched silicone rubber film is much smaller than that of Ag NWs printed on the silicone rubber film without pre-stretching. At pre-stretching 30%, R_stretched_/R_released_ is the minimum at strain 10 and 20%. For Ag NWs electrode printed on the pre-stretched silicone rubber at 30%, R_stretched_/R_released_ at 10%, and 20%, strain was 1.05, and 1.57, respectively, which are significantly less than about 7 for Ag NWs at the 10% strain without pre-stretched substrate. It is thought that the number of contact points with Ag NWs at the stretched state is almost the same as that of a contact point with Ag NWs at released state due to the reversible sliding of Ag NWs. As pre-stretching is above 30%, crack formation on the surface of silicone rubber film caused the Ag NWs to be disconnected. Pre-stretching of silicone rubber film can minimize the resistance difference of stretched and released state.

## Figures and Tables

**Figure 1 nanomaterials-13-00719-f001:**
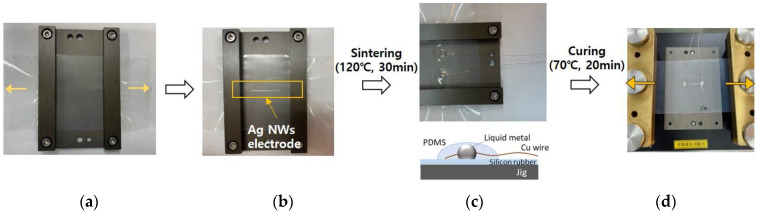
Schematic images of the experimental procedures: (**a**) fixing the pre-stretched silicone rubber film; (**b**) printing, drying, and sintering of Ag NWs line; (**c**) connecting the liquid metal and copper wire; (**d**) cyclic stretching test.

**Figure 2 nanomaterials-13-00719-f002:**
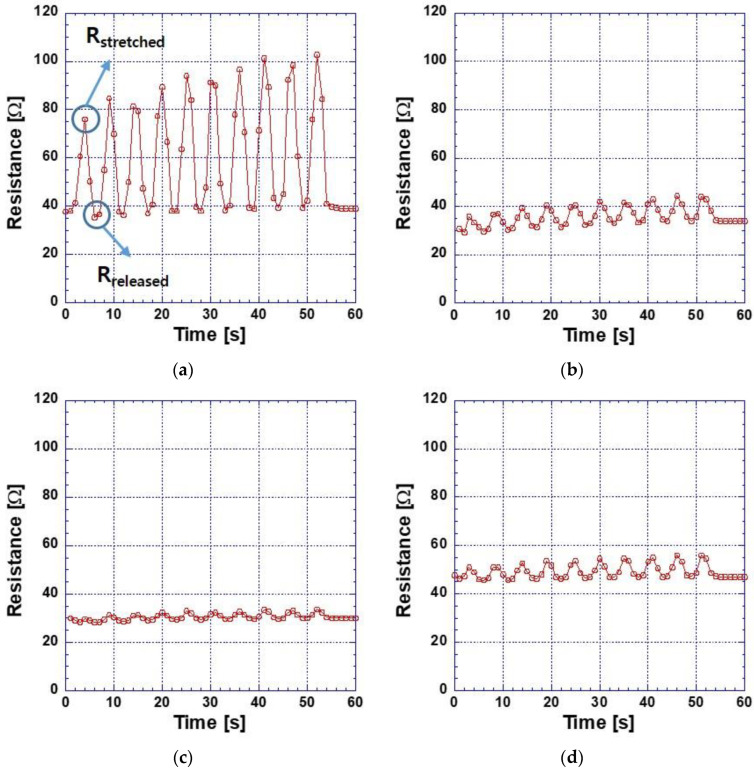
In situ resistance change of Ag NWs electrode on variously pre-stretched silicone rubber film during a cyclic test with 10 times repetition applying 10% strain: (**a**) 10% pre-stretched; (**b**) 20% pre-stretched; (**c**) 30% pre-stretched; and (**d**) 40% pre-stretched.

**Figure 3 nanomaterials-13-00719-f003:**
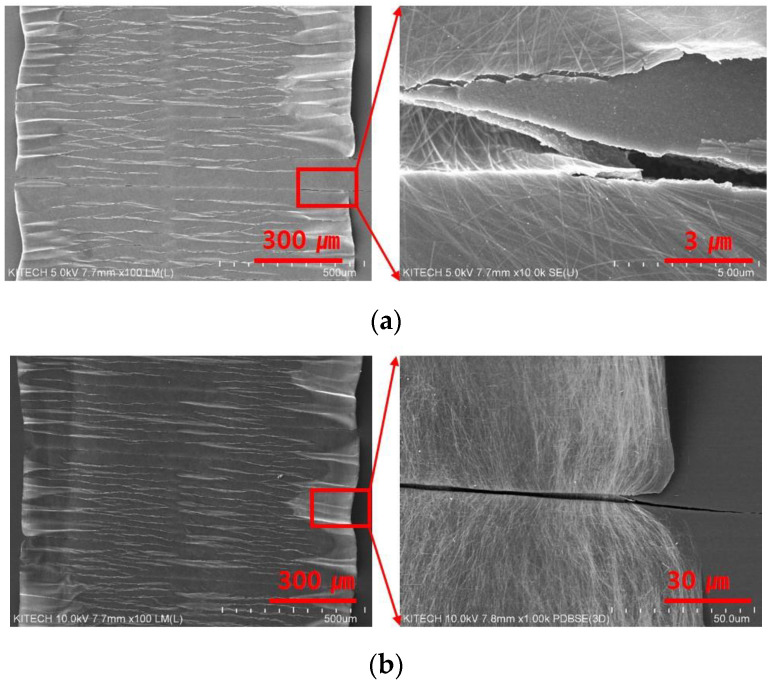
FESEM plan view images of Ag NWs printed on the 40% pre-stretched silicone rubber film: (**a**) 0% strain, (**b**) 10% strain.

**Figure 4 nanomaterials-13-00719-f004:**
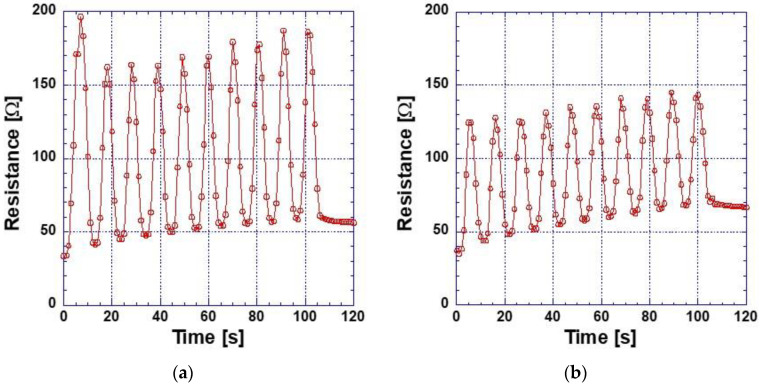
In situ resistance change of Ag NWs electrode on variously pre-stretched silicone rubber film during a cyclic test with 10 times repetition applying 20% strain; (**a**) 10% pre-stretched; (**b**) 20% pre-stretched; (**c**) 30% pre-stretched; and (**d**) 40% pre-stretched.

**Figure 5 nanomaterials-13-00719-f005:**
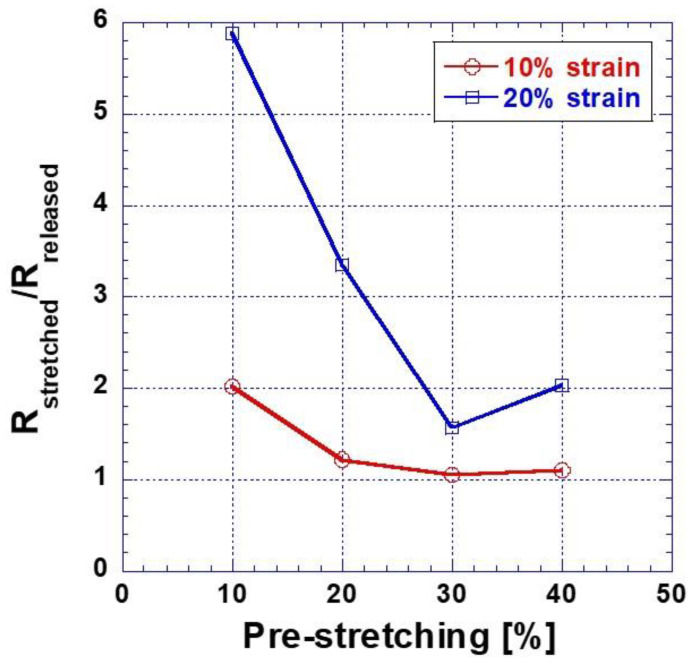
The change of ratio R_stretched_/R_released_ with pre-stretching of silicone rubber (R_stretched_: the resistance of Ag NWs electrode when film was stretched; R_released_: the resistance of Ag NWs electrode when film was released).

**Figure 6 nanomaterials-13-00719-f006:**
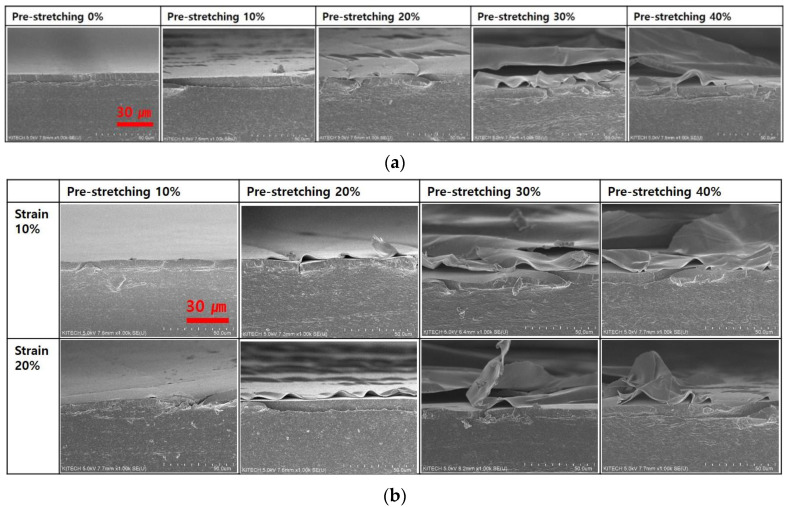
Cross-sectional FESEM images of Ag NWs electrodes with pre-stretching and applying strain: (**a**) 0% strain; (**b**) 10 strain and 20% strain.

**Figure 7 nanomaterials-13-00719-f007:**
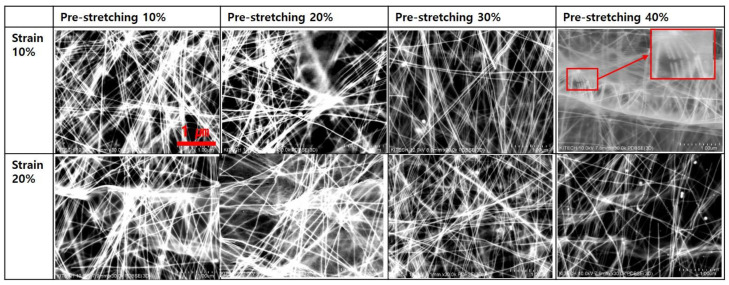
Plan view FESEM images of Ag NWs electrodes with pre-stretching and applying 10 and 20% strain.

**Figure 8 nanomaterials-13-00719-f008:**
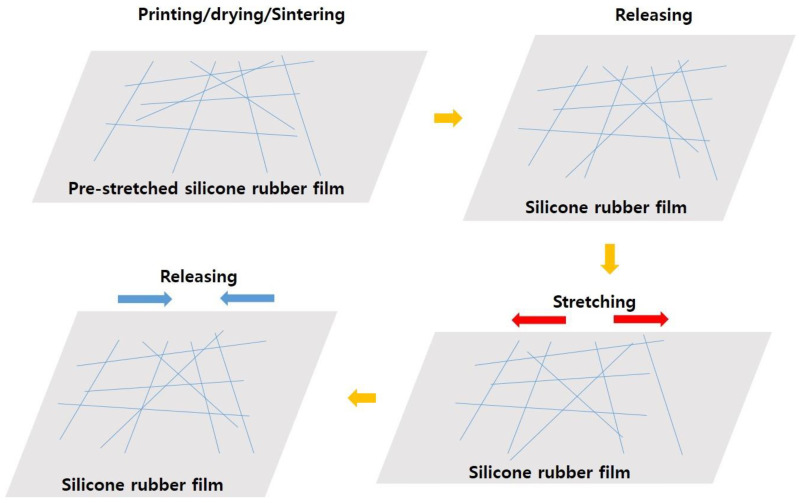
The schematics of Ag NWs sliding applying strain on pre-stretched film.

## Data Availability

Data is contained within the communication material.

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
