# Peer review of "The Effect of Pre-Stretched Substrate on the Electrical Resistance of Printed Ag Nanowires"

_nanomaterials, 2023, doi:10.3390/nano13040719_

Round 1
Reviewer 1 Report
This paper reports a study on the influence of the pre-stretching of the substrate on the electrical resistance variation of printed Ag Nanowires. The pre-stretching is a very commonly used and extensively studied method to create stretchable electronics. This is a widely recognized technique. So, this work is not very innovative and what is the new discovery of this work is not explicitly introduced. I would recommend for publishing only after addressing the following:
1) Please clarify the innovation of this work more explicitly in the “Introduction”.
2) Figure 1: Caption “(b) Connecting the liquid metal and copper wire” should be “(c).
3) What is the silicone rubber film? Is it the same as PDMS? How is the bonding between the PDMS (to fix the liquid metal) and the silicon rubber substrate? Also, how is the bonding between AgNWs and the silicon substrate? Any nanowires are buckled and peeled off after cyclic stretching-releasing? Evidence, please.
4) What is the purpose of sintering? Please specify.
5) Figure 2: It is understandable that the “resistance increase” reduces when the pre-stretching strain increases. However, how to explain that this value increases when the pre-stretching strain increases from 30% to 40%.
6) Also in Figure 2, how to explain the difference in the initial resistance of the sample? Are these the same specimen or different ones?
7) The SEM images shown in Figure 5 are not very clear and can hardly show any differences. A cross-sectional view showing the wavy structure, the AgNWs bonding with silicon substrate is critical. Different structure changes upon stretching may be seen in the cross-sectional view rather than the top view.
8) It is claimed (on page 6) that “These wavy patterns are thought that the surface of silicone rubber film was partially damaged when pre-stretched.” This statement is hard to be believed. What is the failure strain of the silicone rubber substrate? How come it can be damaged at such low pre-stretching strain?
9) The printed AgNWs is in the form of a thin wire. Does the printed pattern also influence the resistance variation upon stretching?
Author Response
Response to Reviewer 1
This paper reports a study on the influence of the pre-stretching of the substrate on the electrical resistance variation of printed Ag Nanowires. The pre-stretching is a very commonly used and extensively studied method to create stretchable electronics. This is a widely recognized technique. So, this work is not very innovative and what is the new discovery of this work is not explicitly introduced. I would recommend for publishing only after addressing the following:
- Please clarify the innovation of this work more explicitly in the “Introduction”.
We appreciate your valuable comment. The pointed out contents was reflected in the manuscript and revised. The revised content was marked by red color (Introduction).
- Figure 1: Caption “(b) Connecting the liquid metal and copper wire” should be “(c).
Thank you. We corrected the your pointing out in the revised manuscript.
- What is the silicone rubber film? Is it the same as PDMS?
Yes, Polydimethylsiloxane (PDMS) is a commercially available physically and chemically stable silicone rubber [1]
How is the bonding between the PDMS (to fix the liquid metal) and the silicon rubber substrate?
During stretching-releasing cyclic test, PDMS was not peeled off from silicone rubber film even at 40% strain.
Also, how is the bonding between Ag NWs and the silicon substrate?
In our previous paper [2], the adhesion of Ag NWs on silicone rubber film was measured using tape test (ASTM D 3359). As shown in Review Figure 1, Ag NWs on silicone rubber film was 1B. The removed area was 37%. Adhesion of Ag NWs on silicone rubber film was not good.
Review Figure 1. Image after tape test (ASTM D 3359) of Silicone rubber film [reference 2]
Any nanowires are buckled and peeled off after cyclic stretching-releasing? Evidence, please.
(a)
(b)
Review Figure 2. FESEM cross-sectional mages of Ag NWs electrode with pre-stretching and applying strain. (a) 0% strain (b) 10% and 20% strain
Review Figure 2(a) shows the FESEM cross-sectional images of Ag NWs electrode with the pre-stretching silicon rubber before applying cyclic stretching and releasing test. From 30% pre-stretching, Ag NWs electrode began to be buckled.
As shown in Review Figure 2(b), Ag NWs electrode printed on 10% pre-stretched silicone rubber film was buckled a little as 10 and 20% cyclic stretching and releasing test. Ag NWs electrode printed on 20% pre-stretched silicone rubber film was buckled. Ag NWs electrode printed on above 30% pre-stretched silicone rubber film was buckled and peeled off.
- What is the purpose of sintering? Please specify.
Generally, sintering is a process of applying temperature and pressure to particles having large specific surface area to make dense lump. Metal nano material ink was separated by the solvent and surfactant to prevent the nozzle clogging caused by agglomeration due to van der Waals force. The separated nano material inks by solvent and surfactant have high initial resistance due to the solvent and surfactant. The connection between nano material is increased and the electrical conductivity is improved by removing the solvent and the surfactant through a sintering process.
In our previous paper [2], we tested the resistance change of the printed siver nanowire with sintering temperature. Review Figure 3 shows the electrical resistance change of Ag NWs with sintering temperature. Electrical resistance was 1.6 kΩ at the room temperature (25 ◦C), and decreased until 100 ◦C. At 100 ◦C, electrical resistance was about two times lower than at 25 ◦C. Above 100 ◦C, electrical resistance was slightly increased until 170 ◦C. Above 100 ◦C, electrical resistance was similar. But standard deviation of electrical resistance at 120 ◦C was small. So, sintering condition was decided as 120 ◦C for 30 min at the experiment.
Review Figure 3. Electrical resistances of Ag NWs electrode with sintering temperature
- Figure 2: It is understandable that the “resistance increase” reduces when the pre-stretching strain increases. However, how to explain that this value increases when the pre-stretching strain increases from 30% to 40%.
- (b)
Review Figure 4. FESEM plan view images of Ag NWs printed on the 30% pre-stretched silicone rubber film. (a) 0% strain, (b) 10% strain
(a)
(b)
Review Figure 5. FESEM plan view images of Ag NWs printed on the 40% pre-stretched silicone rubber film. (a) 0% strain, (b) 10% strain
I appreciate your valuable comment. As shown in Review Figure 4, in the Ag NWs electrode on 30 % pre-stretched silicone rubber film, Ag NWs was buckled and Ag NWs was not damaged. However, as shown in Review Figure 5, Ag NWs electrode printed on the 40% pre-stretched silicone rubber film was broken due to the crack of silicone rubber film at 0% strain. At 10% strain (Review Figure 5(b)), crack was propagated during cyclic stretching and releasing test. So, it is thought that electrical resistance of Ag NWs printed on the silicone rubber film was increased compared with that of Ag NWs printed on the 30% pre-stretched silicone rubber film. This content was included in the revised manuscript to Added Figure 6.
- Also in Figure 2, how to explain the difference in the initial resistance of the sample? Are these the same specimen or different ones?
Thank you for your comment. In the printed metal electrode, initial resistance can be different according to printing condition, such as temperature humidity, ink condition etc. In our previous paper [2], we printed the Ag NWs electrode using the same dispenser and ink. However, after sintering, initial resistance was not same. The standard deviation of initial resistance was about 20% before stretching test (shown as the error bar in Review Figure 6).
In this manuscript, Ag NWs electrode was printed the same day until 30% pre-stretching. Average initial resistance of Ag NWs electrode was 33. 8 Ω and standard deviation was approximately 3 Ω. After a few days, Ag NWs electrode on the 40% pre-stretched silicone rubber film was printed. After sintering, average initial resistance was 52.1 Ω and standard deviation was 5.1 Ω. % standard deviation of initial resistance of Ag NWs electrode on the 40% pre-stretched silicone rubber film was similar to that of Ag NWs electrode on the 10, 20, and 30% pre-stretched silicone rubber film. As shown in Review Figure 5(a), due to the 40 % pre-stretching, it is thought that the higher initial resistance of Ag NWs printed on the 40% pre-stretched silicone rubber film resulted from the break of Ag NWs electrode due to the crack formation of silicone rubber film.
Review Figure 6. Resistance change of Ag NWs line on silicone rubber film after various strains
- The SEM images shown in Figure 5 are not very clear and can hardly show any differences. A cross-sectional view showing the wavy structure, the Ag NWs bonding with silicon substrate is critical. Different structure changes upon stretching may be seen in the cross-sectional view rather than the top view.
I appreciate your valuable comment. We performed cross-sectional FESEM observation. We could observe the wavy structure of Ag NWs electrode. Cross-sectional FESEM (Revised Figure 5) and plan view images (Added Figure 7) was reflected and the explanation was added in the revised manuscript.
- It is claimed (on page 6) that “These wavy patterns are thought that the surface of silicone rubber film was partially damaged when pre-stretched.” This statement is hard to be believed. What is the failure strain of the silicone rubber substrate? How come it can be damaged at such low pre-stretching strain?
I appreciate your valuable comment. The silicone rubber film that we used in this experiment was purchased from the trade agent (ALPHAFLON, Korea). Product name was silicone rubber 0.3T. They did not tell the manufacturer and model number. They only provide the test report about silicone rubber film. Tensile strength, elongation percentage and hardness (Hs) are 8.6 MPa, 410%, and 60 respectively. As shown in Review Figure 5(a), due to 40% pre-stretching of silicone rubber film, the partial damage of Ag electrode and the surface of silicone rubber film was observed. And we observed the wavy patterns (Review Figure 2). Because the surface of original silicone rubber film was hydrophobic, we did UV/O3 treatment to the surface of silicone rubber film to hydrophilic surface in order to print the Ag NWs ink on it well. The reason why crack occurred on the silicone rubber surface at 40% pre-stretching strain is due to the formation of silicon oxide (SiOx) on the silicone rubber surface by UV/O3 treatment before printing the Ag NWs electrode [3]. This content was reflected in the revised manuscript.
- The printed Ag NWs is in the form of a thin wire. Does the printed pattern also influence the resistance variation upon stretching?
I appreciate your comment. The printed pattern can influence the resistance variation upon stretching in the case which line width is small. Pattern dimension of our Ag NWs electrode is 20 mm in length and 1.2 mm in width. The width of Ag NWs line is big compared with that of other printed lines. It is thought that resistance variation upon stretching may not be influenced by our pattern dimension.
Review references
[1] J. C. Lötters, W. Olthuis, P.H. Veltink, P. Bergveld, The mechanical properties of the rubber elastic polymer polydimethylsiloxane for sensor applications, J. Micromech. Microeng., 1997, 7, 145
[2] Y. Moon, K. Kang, Strain-induced alignment of printed silver nanowires for stretchable electrodes, Flex. Print. Electron., 2022, 7, 024003
[3] Roth J, Albrecht V, Nitschke M, Bellmann C, Simon F, Zschoche S, Michel S, Luhmann C, Grundke K and Voit B, 2008, Surface functionalization of silicone rubber for permanent adhesion improvement, Langmuir, 24, 12603–11
Reviewer 2 Report
Dear authors,
the FESEM images in Figure 5 have picture captions, which are not readable. It would be important to understand the size of the features.
Moreover the FESEM images should support your explanation of the mechanism during stretching and release shown in Figure 6. It would be helpful to indicate these in the images of Fig. 5.
With kind regards
Author Response
Response to reviewer 2
The FESEM images in Figure 5 have picture captions, which are not readable. It would be important to understand the size of the features.
Moreover the FESEM images should support your explanation of the mechanism during stretching and release shown in Figure 6. It would be helpful to indicate these in the images of Fig. 5.
I appreciate your valuable comment. We performed cross-sectional FESEM observation. We could observe the wavy structure of Ag NWs electrode. Cross-sectional FESEM and plan view images was reflected in the revised manuscript. Figure 5 was changed to cross-sectional FESEM images (Revised Figure 5) and explanation was added. And Figure 6 and Figure 7 were added as Added Figure 6 and 7. According to your comment, we used more distinct images. Scale bar was also added in the images.

Reviewer 3 Report
This paper reports on the effect of the pre-stretched silicon rubber film substrate on electrical resistance of the printed Ag nanowires (NWs). The main result shows that the difference of Rstretched and Rreleased of Ag NWs printed on the pre-stretched silicone rubber film is much smaller than that of Ag NWs printed on the silicone rubber film without pre-stretching. Therefore, pre-stretching of the substrate can potentially minimize the difference in the resistance for the stretched and released states. This may be important for applications of these Ag NWs in electronics on flexible substrates. The paper is well written and nicely illustrated, with relevant references. Overall, the results are new and should be of interest for a wide readership. Therefore, I recommend publication as is.
Author Response
Response to reviewer 3
This paper reports on the effect of the pre-stretched silicon rubber film substrate on electrical resistance of the printed Ag nanowires (NWs). The main result shows that the difference of Rstretched and Rreleased of Ag NWs printed on the pre-stretched silicone rubber film is much smaller than that of Ag NWs printed on the silicone rubber film without pre-stretching. Therefore, pre-stretching of the substrate can potentially minimize the difference in the resistance for the stretched and released states. This may be important for applications of these Ag NWs in electronics on flexible substrates. The paper is well written and nicely illustrated, with relevant references. Overall, the results are new and should be of interest for a wide readership. Therefore, I recommend publication as is.
Thank you very much for your kind comment.

Round 2
Reviewer 1 Report
The authors have addressed my comments.